# Is Right Angular Gyrus Involved in the Metric Component of the Mental Body Representation in Touch and Vision? A tDCS Study

**DOI:** 10.3390/brainsci11030284

**Published:** 2021-02-25

**Authors:** Grazia Fernanda Spitoni, Giorgio Pireddu, Valerio Zanellati, Beatrice Dionisi, Gaspare Galati, Luigi Pizzamiglio

**Affiliations:** 1Cognitive and Motor Rehabilitation and Neuroimaging Unit, IRCCS-Santa Lucia Foundation, 00179 Rome, Italy; gaspare.galati@uniroma1.it; 2Department of Dynamic and Clinical Psychology, and Health Studies, Sapienza University of Rome, 00185 Rome, Italy; 3Italian Association of Specialists in Neuropsychology, 00179 Rome, Italy; giorgio.pireddu@gmail.com; 4Sapienza University of Rome, 00185 Rome, Italy; valeriozanellati@gmail.com (V.Z.); luigi.pizzamiglio@uniroma1.it (L.P.); 5Italian Society of Psychoanalysis of Relation SIPRe, 00183 Rome, Italy; dionisibeatrice1@gmail.com; 6Department of Psychology, Sapienza University of Rome, 00185 Rome, Italy

**Keywords:** body representation, angular gyrus, tDCS, IPL, distance discrimination

## Abstract

Several studies have found in the sense of touch a good sensory modality by which to study body representation. Here, we address the “metric component of body representation”, a specific function developed to process the discrimination of tactile distances on the body. The literature suggests the involvement of the right angular gyrus (rAG) in processing the tactile metricity on the body. The question of this study is the following: is the rAG also responsible for the visual metric component of body representation? We used tDCS (anodal and sham) in 20 subjects who were administered an on-body distance discrimination task with both tactile and visual stimuli. They were also asked to perform the same task in a near-body condition. The results allow us to confirm the role of rAG in the estimation of tactile distances. Further, we also showed that rAG might be involved in the discrimination of distances on the body not only in tactile but also in visual modality. Finally, based on the significant effects of anodal stimulation even in a near-body visual discrimination task, we proposed a higher-order function of the AG in terms of a supramodal comparator of quantities.

## 1. Introduction

From early studies to today, body perception and representation have never ceased to stimulate questions, hypotheses, and research in both the representation of the body as a whole and the representation of specific body parts.

We know that most studies have used vision as the preferred modality for the study of body representations. Nevertheless, in the last decade, several works have found in the sense of touch another good modality by which to study body representation [1,2,3]. The most interesting aspect of the use of touch for investigating body representation lies in the possibility of analyzing a series of perceptual phenomena indirectly. In other words, in touch studies, body representation is inferred on the basis of tactile tasks that do not directly involve the request for body evaluation. We owe to the model proposed by Serino and Haggard (2010) [4] one of the first attempts to describe the various anatomo-functional steps that, starting from a touch, lead to a body representation. According to the authors, in order to reach a representation of one’s own body from an object touching the skin, four processes are required: (1) the physical body arranges tactile sensations, (2) tactile sensations contribute to the development of a mental body representation, (3) mental body representation decisively influences primary tactile processing, and (4) it mediates the shaping of object representation from primary tactile sensations. This model suggests that the brain computes several sources of information (i.e., tactile, visual, and proprioceptive) to scale information about skin contact in relation to the perceived size of the body part tactilely stimulated. In other words, in order to make a judgment about the size of an object that touches our skin, we need to recall the representation of the part of the body touched and rescale the size of the object relative to it. We must, therefore, represent the part of the body touched.

The dimension that we would like to address here is what has been called “the metric component of body representation”. In a previous work [5], we used the term “Metric Component of the Body Representation” (MCBR) to describe a selective dimension of body representation used to process the discrimination of tactile distance [6]. Using a distance discrimination task, we compared the activation of brain areas when healthy subjects had to compare the pressure versus the distance of tactile stimuli applied to different body surfaces. Both tasks (pressure–distance) bilaterally activated parietal and frontal areas. However, the metric judgment (namely, the distance task) on the body surface activated the angular gyrus (AG) and the TPOJ in the right hemisphere. The involvement of different brain areas in the two tactile tasks was interpreted as being due to the need for using a MCBR only in the distance task. Another study [7] confirmed the selective involvement of the right posterior inferior parietal lobe by asking to compare the distance between two touches across different body parts (forearm versus sternum): the performance (accuracy and response times) after a tDCS stimulation on the right and left angular gyrus showed a significant perturbation in the distance task only for the right tDCS. Both studies showed a lateralized right posterior parietal involvement in the metric spatial evaluation of the body.

Both studies seem to suggest that in order to perform a distance comparison between two touched points across different body parts we should hypothesize a touch sensitive system (i.e., a mental body representation), which represents the state of the body rather than the physical features of the touching stimuli.

In this framework, MCBR seems to represent a specific function of a mental body representation. We consider the following question: is MCBR specific to tactile modality? Is the right angular gyrus also responsible for the visual metric component of body representation?

Although the literature on the function of vision of body representation and the space around the body is extremely rich, at present, the visual metric component of body representation does not appear to be investigated.

To this end, we performed a tDCS study in which the right angular gyrus (rAG) was perturbed, while healthy participants performed a visual distance discrimination task on and off the body. The variables analyzed were response time and accuracy. We hypothesize that if the rAG has a specific modulatory function for the metric representation of sizes only in the tactile modality, the perturbation by tDCS should not affect the performance of the same task administered in the visual modality. Alternatively, a modulatory effect of tDCS on the visual distance discrimination task on the body (and not out of the body) might suggest an a-modal role of the rAG. As this study can be conceptualized as a continuum with the other two previous studies, we used anodal and sham stimulation in all conditions.

## 2. Materials and Methods

### 2.1. Subjects

Twenty (9 men) healthy volunteers participated in the study. One was excluded because his behavioral performance was at chance level in the distance task. The remaining nineteen subjects aged from 21 to 27 years (mean = 24.2; SD = 2.6) and were all right-handed, as assessed by a modified version of the Edinburgh Inventory [8,9]). All of them had no history of neurological and psychiatric diseases, were in good health, and were not on medication. The experiment was conducted in accordance with the Code of Ethics of the World Medical Association (Declaration of Helsinki) [10]. All participants provided written informed consent, and the study was approved by the Committee on Research Ethics of IRCCS Santa Lucia Foundation. Participants received a voucher to spend in a bookstore.

### 2.2. Method

All participants underwent the same experimental protocol, consisting of two experiments. The total duration of the whole protocol was about two and a half hours divided into three days. The administration of the conditions was balanced across subjects.

In the first study, we investigated whether anodal perturbation of the right angular gyrus (rAG) had a modulatory effect on a task of visual distances performed on the body and surrounding space.

The second was a confirmatory experiment. The goal was to test and possibly replicate previous findings on the effects of anodal stimulation of the rAG on a tactile distance task.

Due to technical differences between the two studies, inferential analyses were conducted separately.

#### 2.2.1. Visual Experiment

##### Tasks and Stimuli

Each subject underwent two visual conditions, namely, a distance discrimination task on the body (body task) and distance discrimination task on the desk (desk task).

In the body task, the subject was seated and placed his or her right arm on a desk; the left arm was placed under the desk in a position considered by the subject as comfortable. In the desk task, the subject remained in the same position but, unlike the previous task, the right arm was moved under the desk, lying on the right thigh (basically in the same position as in the body task, but under the desk).

Visual stimuli were delivered by a laser projector able to produce two simultaneous dots (1.5 mm diameter) on a given surface. Accordingly, the visual stimuli consisted of pairs of visual dots separated by a variable distance. In the body task, we stimulated the right forearm and sequentially the right hand (ISI = 100 ms) (dorsum), whereas in the desk task, the stimuli were projected onto the part of the desk previously occupied by the subject’s arm showing an “up” or “down” area (see Figure 1).

Participants were instructed to say in which condition (forearm/hand, or up/down) the distance between the two dots was larger. In each trial, either the forearm or the hand and the up or the down stimulus was of fixed dimensions (distance = 4 cm), and the other randomly changed. The latter was divided into 4 levels of difficulty, depending on the difference in amplitude present between the first and second stimuli projected in sequence. The levels were labeled “easy” (±3 mm), “medium difficulty” (±2 mm), “difficult” (±1 mm), and “no difference” (0 mm). These distances were fine-tuned from a previous study designed to construct a psychophysical curve to account for different levels of difficulty (the description and details of the stimulus calibration can be found in the Appendix A).

All subjects received 20 trials for each difficulty level (0, ±1, ±2, ±3) in each condition (body/desk) for each session (sham/anodic). Thus, we obtained 140 trials for each condition, for a total of 560 trials. Stimuli were balanced per comparison, and accuracy and vocal reaction times were recorded.

##### tDCS Protocol

Anodal stimulation at 2 mA for 15 min was used. The current was produced by a battery-driven, constant-current stimulator (Rolf Schneider Electronics, Germany) diffused by two saline-soaked surface sponge electrodes (7 × 4.5 cm). Previous studies have shown that at this intensity, the procedure falls well within the criteria proposed by the safety protocols for the use of tDCS [11,12].

In the stimulation sessions, current increased in a ramp-like fashion from 0 to 2 mA at the onset of stimulation, eliciting a transient tingling sensation on the scalp [12]. After 15 min, the current gradually decreased until it returned to zero in 30 s.

In the sham stimulation, the current began to gradually increase from 0 to 2 mA in 30 s and was then turned off, stopping the stimulation.

In both conditions, the tasks (body and desk) were performed simultaneously with the tDCS stimulation (online performance).

Montage. The anode was applied on the right parietal lobe (anode at the angular gyrus-CP4-International System 10–20), while the cathode at the left frontoparietal site (Fp1). This electrodes displacement ensures adequate stimulation in which most of the current reaches the target parietal region (namely rAG), and it respects the safety rules required for the use of tDCS. Participants received anodal or sham stimulation in both the body and desk task; to avoid the carry-over effect of anodal stimulation, tasks and stimulations were performed over two days and in a balanced order across subjects.

Figure 2 summarizes the experimental paradigm.

##### Analyses

Both response times and accuracy were analyzed across two sets of Analyses of Variance (ANOVA). Response time refers to the time between perceiving a stimulus and responding to a specific question about it. It is usually measured in milliseconds. Accuracy refers to % of correct responses. In either case, we used a completely repeated ANOVA design with task (body and desk), stimulation type (anodal or sham), and difficulty (easy, medium, difficult) as within-subject factors. Bonferroni post hoc tests corrected for multiple comparisons (*p* < 0.05) were conducted.

#### 2.2.2. Tactile Experiment

##### Tasks and Stimuli

In the tactile experiment, we used the same protocol already used in a previous experiment [7], and the tactile stimuli were delivered by a set of mechanical solenoids interfaced with a computer device. Participants received anodal or sham conditions in the same day with an interval of at least 6 h between the two sessions; the use of a long interval allowed us to avoid the carry-over effect of the anodal stimulation and to balance the administration across participants. As the nature of the touch does not allow us to test a “non-body” surface, in this experiment, the desk task was not performed. Additionally, since the tactile stimulator does not allow for a large number of comparisons, the difficulty of the task was limited to easy and difficult comparisons. This means that unlike the visual task, the difficulty of the trials was operationalized with only two levels instead of three.

##### tDCS Protocol

In this experiment, we used the same montage adopted for the visual study, and the stimulation procedures were the same. In this case, performance of the tactile task was also carried out simultaneously with tDCS stimulation.

##### Analyses

Response times and accuracy were analyzed. To this end, two separate 2 × 2 repeated ANOVAs were conducted, with stimulation type (anodal or sham) and difficulty (easy or difficult) as within-subject factors.

## 3. Results

### 3.1. Experiment 1

A significant main effect of difficulty emerged [F _(1, 2509)_ = 9.6; *p* < 0.001; η2 = 0.06], with faster responses to easy (596.5 ms) than medium (676.7 ms) and difficult judgments (722.1 ms) (see also Appendix A).

Significant main effects of the task [F_(1, 2509)_ = 5.52; *p* < 0.01; η^2^ = 0.04] and stimulation [F_(1,2509)_ = 4.34; *p* < 0.05; η^2^ = 0.01] also emerged, with a reduction in Response Times (RTs) on the desk and in the anodal stimulation, respectively. Finally, a significant difficulty by task interaction [F_(2,2509)_ = 11.76; *p* < 0.001; η^2^ = 0.08] emerged with faster RTs in the easier trials on the desk (Figure 3). The remaining two-way and three-way interactions were not significant [F_(2,2509)_ = 3.09; *p* = 0.079, ns; and F_(2,2509)_ = 0.22; *p* = 0.88, ns, respectively].

With regard to accuracy, a significant main effect of difficulty emerged [F_(2, 2994)_ = 21.3; *p* < 0.001; η^2^ = 0.07], with a higher rate of accuracy in the easy (78%) than in the medium (69%) and difficult stimuli (67%). We also found a significant effect of the task [F_(2, 2994)_ = 11.1; *p* < 0.01; η^2^ = 0.06] with performance on the desk (82%), more accurate than on the body (76%). The effects of the stimulation [F_(2, 2994)_ = 2.72; *p* = 0.09] and interactions were not significant [F_(2, 2994)_ = 3.49; *p* = 0.07].

In sum, results from the first experiment indicate that the visual distance discrimination task on the desk is easier than that on the body (faster RTs and higher accuracy). Additionally, stimulation of the rAG affects vRTs both in the body and desk tasks.

### 3.2. Experiment 2

Two separate 2X2 repeated measures of ANOVA over RTs showed the main effects of difficulty [*F*_(1, 41)_ = 22.1; *p* < 0.001; η^2^ = 0.06] and stimulation [*F*_(1, 41)_ = 17.1; *p* < 0.01; η^2^ = 0.03]. Subjects were overall faster in the easy trials (881 vs. 896 ms) and during anodal stimulation (747 vs. 842 ms) (Figure 4). The interaction was not significant [*F*_(1, 41)_ = 8.2; *p* = 0.09, ns]

The same ANOVA model of accuracy (%) showed the main effect of difficulty [*F*_(1, 41)_ = 21.2; *p* < 0.01; η^2^ = 0.04], with better performance in easy trials (80% vs. 72%).

## 4. Discussion

The metric component of body representation allows people to perceive the size of an object on the body; studies on touch have shown that the rAG may be involved in this function.

The first result we would like to discuss is related to the replication of this evidence. In fact, similarly to a previous study [7], experiment 2 showed that anodal perturbation of the rAG is able to modulate a tactile distance discrimination task on the arm. This result is particularly interesting because, despite the fact that in the two experiments the electrical stimulation took place with different methods (off-line in Spitoni et al., 2013, and on-line in the present study) [7], the results converge in showing a reduction in vocal response times when anodal stimulation is given to the rAG. It is also worth mentioning that the data on accuracy were replicated and appear to be unaffected by anodal stimulation. This latter finding supports a number of studies (for review see 35), which suggest the efficacy of a tDCS in reducing response times in the absence of an improved accuracy in cognitive tasks [13].

The main aim of the present study, besides merely replicating the effect of anodal electrical stimulation on tactile distance processing, was to understand whether the rAG is implicated in processing distances on the body when given in another sensory modality. Our results clearly show that the discrimination of visual distances on the body is affected by anodal rAG stimulation; the results of the first experiment show that, during anodal stimulation, vocal response times in the visual discrimination task get significantly faster under rAG stimulation, suggesting that, as with tactile distance judgments on the body, rAG is also involved for visual ones. Moreover, accuracy is not affected by electrical perturbation in this case either. With respect to our hypothesis, it thus appears that the process of discriminating distances on the body can be, at least in part, hosted by the rAG.

Note that, for methodological reasons, the results of the two experiments are not directly comparable to each other. Indeed, the difficulty levels of the tactile task were less (2) than the visual ones (3) because of technical limitations in the solenoid arrangement and, more importantly, the tactile condition cannot include an off-body condition. However, a qualitative comparison shows that the effect of rAG stimulation on the distance perception task (i.e., a reduction in vocal response times in the absence of any accuracy effect) is the same across the two sensory modalities.

This result, however, must be necessarily discussed with respect to another result that emerged in experiment 1. Indeed, the same tDCS protocol also seems to have a comparable effect in the off-body distance discrimination task (desk task). The desk task was thought of as a way to dissociate the processing of visual distances on-the-body from visual processing of distances per se. The presence of a significant effect of rAG stimulation in the desk task is thus both unexpected and intriguing: although it does not undermine the interpretation of the function of the rAG in the distance discrimination task on the body, it causes us to reconsider it within the broader spectrum of functions this brain region has in perceptual tasks.

We believe there are two possible explanations for this result, which are not mutually exclusive and are at the moment speculative, and may be the basis of further studies on the topic. First, we should think of this inferior parietal area as being involved in the process of discrimination of spatial magnitudes, regardless of the type and modality of the task, in other words, a sort of “supramodal comparator of sizes”. It may seem simplistic that, along with the numerous functions ascribed to the AG [14], we can also include this one. Nevertheless, a similar interpretation was provided by van Kemenade et al. (2017), who proposed that the angular gyrus acts as a supramodal comparator area between perception and interpretation for both unimodal and bimodal action consequences [15].

Furthermore, several studies suggested a strong involvement of the left AG in number processing, such as digit subtraction and number comparison [16,17,18,19]. It has also been proposed that the representations of number and spatial distances are perhaps somehow connected by a partially common neural substrate [20,21]. For example, Pinel and colleagues (2004) used fMRI to investigate the brain regions involved when subjects were asked to compare pairs of stimuli on the basis of number, size, and luminance [22]. They found activations of the inferior parietal cortex during number and size comparisons. To support this finding, a rTMS study showed that the stimulation of the IPS can disrupt both numerical and length judgment task stimulation [23].

To conclude, we can hypothesize that the effect of anodal tDCS also in the off-body task can be interpreted in light of a possible supramodal function of the rAG in processing comparisons of quantities. Furthermore, the involvement of the right and not left AG is compatible with the spatial nature of the task.

The second possibility is that the desk task does not completely eliminate the body-related nature of the proposed spatial comparison, because it is executed in the immediate vicinity of the body. That the visual space around the body is represented, especially in the parietal cortex, in an integrated way with tactile and proprioceptive signals coming from the body, is a well-established finding in cognitive neuroscience, based on a vast amount of data, ranging from psychophysical evidence to single-neuron recordings in behaving animals. In the present study, we chose to present the distance comparison task on the desk to maintain the same physical location of the visual stimuli as in the body task. This necessarily implied that the stimuli were presented close to the body, and we thus cannot exclude that some form of multimodal (visuo-tactile) body representation was implicitly elicited by the request of processing spatial features of peripersonal visual space. It would be interesting to replicate these findings in the so-called “far” space, i.e., asking to judge stimuli outside reaching distance.

Another finding from experiment 1 is that responses on the desk were faster and more accurate than those on the body. With respect to speed, this may be because the body task requires additional processing, such as retrieving an intrinsic property of the stimulus from the contingent properties of its contact with the skin. Regarding the latter interpretation, we could, therefore, hypothesize that only in the body task is a mental body representation recruited. With respect to accuracy, we know that in human life, actions are typically guided by the vision of the external environment. For example, every time we interact with an object, we have to represent its visual characteristics, such as shape, size, orientation, and temperature [24,25]. So, the experience of evaluating quantities outside of our body is certainly more frequent than evaluating distances on the body. Therefore, we could suggest that the performance on the desk is faster and more accurate because of the effect of “practice and experience”.

To briefly summarize, in this study, the role of rAG in the estimation of tactile distances was confirmed; this evidence corroborates the findings of other studies that have used a similar paradigm for the study of body representation both in patients and healthy subjects [26,27,28,29]. We also showed, for the first time, that rAG might be involved in discrimination of distances on the body not only in tactile but also in visual modality. Finally, based on significant effects of anodal stimulation even in a “near the body visual discrimination task” (desk), we proposed a higher-order function of the AG in terms of a supramodal comparator of quantities.

The limits of our study should be considered.

It is clear that the first one concerns the low spatial focality of tDCS. In fact, we know that a part of the current delivered by the electrodes spreads in brain areas adjacent and under the target area [30,31]; therefore, the interpretation of data must consider the possible effect of stimulation on the latter areas. However, it is known that the greater part of the delivered current affects the cortical area perpendicular to the active electrode [30,32,33,34,35,36]. This evidence allows more confidence in the validity of our data.

Another limitation concerns the absence of stimulation of the left AG. Because this is the first study to investigate visual distance discrimination on the body, it would have been suitable to stimulate the left hemisphere as well. We chose not to do so because, to avoid individual variability, we wanted to perform the study completely within subjects. In order to balance the conditions (sham/anodal), the duration of the experiment was already very long (and the addition of stimulation to the left hemisphere would have required it to be twice as long). Moreover, in previous studies, a specific effect of the right AG and not of the left AG on the discrimination of tactile distances has been shown [7]. Obviously, in light of a possible supramodal role of the AG, it would be interesting to repeat the experiment stimulating also the left AG.

Finally, it is possible that a control task on the body and not in space (e.g., discrimination of visual distances made with different colors and asking subjects to discriminate which of the two stimuli was darker) would have allowed further interpretability of the data.

In conclusion, this study confirmed the role of the rAG in the processing of on-body and off-body distances, providing new evidence on both visual and tactile assessment.

Further studies, perhaps using different techniques, will certainly be needed to confirm these data, providing greater consistency to our evidence.

## Figures and Tables

**Figure 1 brainsci-11-00284-f001:**
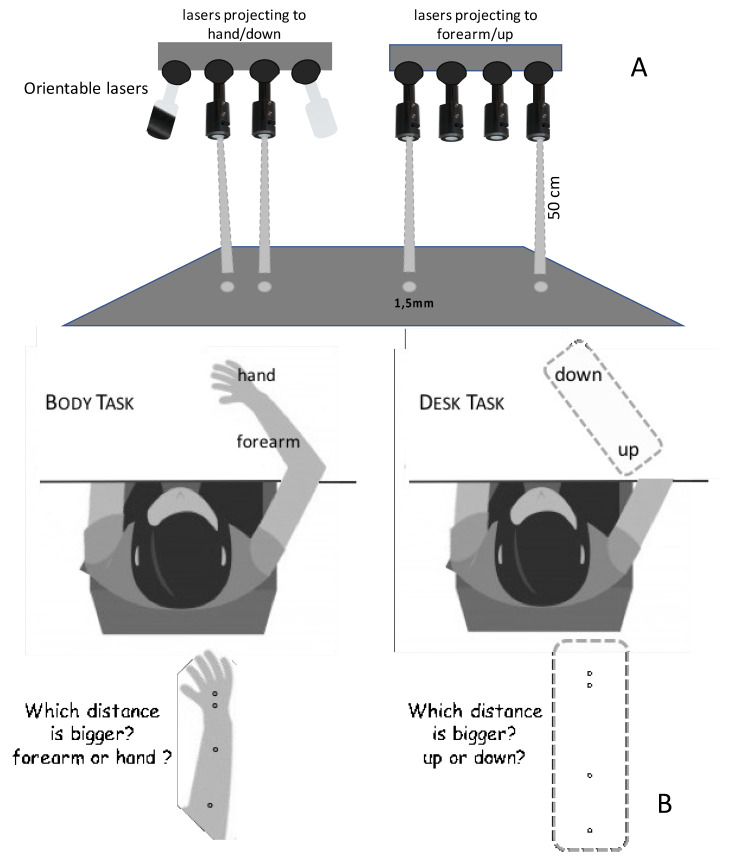
Experimental setup. (**A**) Light dots are produced by a laser system that projects pairs of light dots (yellow light) onto the body (forearm and hand) and desk (up or down). Lasers are orientable so as to provide the same “distances” for each subject regardless of the size of their arms. (**B**) In the body task, the subject receives the stimuli on the forearm and on the hand (ISI = 100 ms); in the desk task, the stimuli are projected onto the two parts of the desk previously occupied by the forearm and the hand. In both cases, the subject is asked to evaluate which of the two stimuli is bigger.

**Figure 2 brainsci-11-00284-f002:**
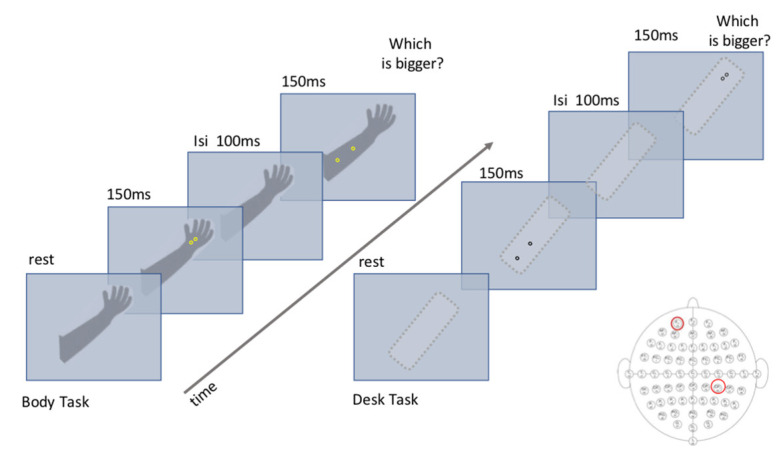
Experimental paradigm of the body task and desk task. In each trial, the first stimulus could be projected independently on the hand or forearm in the body task and on the up or down side in the desk task. In each trial, one of the two stimuli was a fixed reference of 4 cm.

**Figure 3 brainsci-11-00284-f003:**
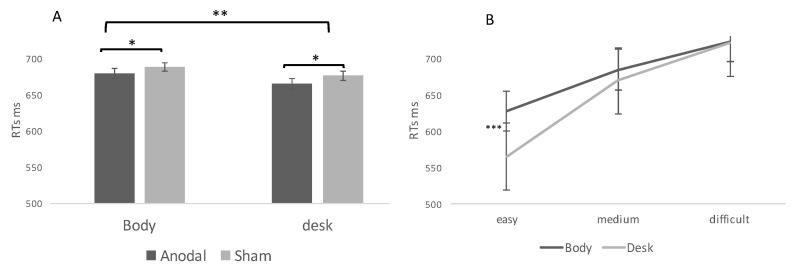
Results of experiment 1. (**A**) Main effects of task and stimulation. (**B**) Interaction task by difficulty. Note: * = *p* < 0.05; ** = *p* < 0.01; *** = *p* < 0.001.

**Figure 4 brainsci-11-00284-f004:**
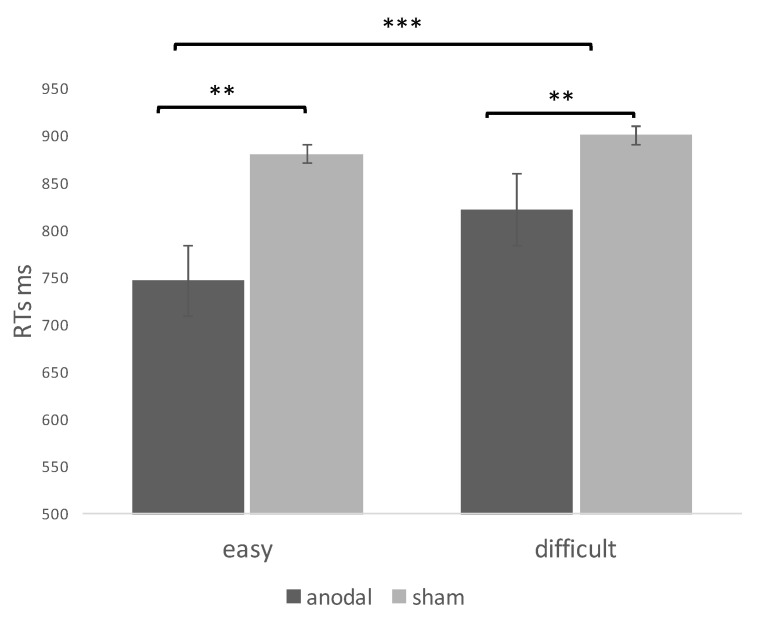
Results of experiment 2. Main effects of difficulty and stimulation. Note: ** = *p* < 0.01; *** = *p* < 0.001.

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
