# Peer review of "Is Right Angular Gyrus Involved in the Metric Component of the Mental Body Representation in Touch and Vision? A tDCS Study"

_brainsci, 2021, doi:10.3390/brainsci11030284_

Round 1

Reviewer 1 Report

This is a novel and well-conducted study. 

Below are my few comments:

  1. There was a stimulation effect on reaction time, but not on accuracy. What does it say about the comprehensive role of rAG in the discrimination of distances on the body in visual modality? Please discuss.
  2. It is important to report all statistical effects (main and interaction) along with F and P values. Also, the authors must report a measure of effect size. 
  3. The authors must report error bars in their figures. They are missing from all plots. 
  4. Figures - Please use similar y-axis scale for sub-plots within a figure.
  5. Please clearly define reaction time.
  6. Please justify the stimulation montage used in this study.
  7. Did the authors randomize the conditions to rule out any order effect?

Author Response

The authors would like to thank the 3 reviewers for the careful analysis of the draft and the time spent. Each point raised is appropriate and has helped to improve the manuscript. We hope we have interpreted the questions correctly

Below the replies provided point by point. (when provided, the mention to the number of lines refers to the revised version of the manuscript and NOT to the first version)REV#1

This is a novel and well-conducted study. 

Below are my few comments:

  1. There was a stimulation effect on reaction time, but not on accuracy. What does it say about the comprehensive role of rAG in the discrimination of distances on the body in visual modality? Please discuss.

REPLY: The impact of tdcs on reaction time but not on accuracy is an effect that has been evidenced in several studies. A meta-analysis by Dedoncker et al. (2016), described that following a-tDCS over anterior regions, healthy subjects responded faster, while neuropsychiatric patients responded more accurate. Also, in a previous study by our group (spitoni et al., 2013), we found that off-line a- tDCS over right angular gyrus improved vocal RTs but not accuracy. The reasons for such dissociation can only be hypothesized. On the one hand it is possible to imagine that some parameters such as the duration and, above all, the density of stimulation may play a role in modulating the facilitative effect of a-tDCS on accuracy.

From another perspective, more personal and speculative, it seems that in cognitive tasks, the effect of tDCS on accuracy emerges especially when the task is very difficult. In this framework it is not surprising that in the meta-analysis of Dedoncker and collaborators, an effect on accuracy emerges only in patients who, evidently, because of cognitive deficits, perceive tasks as more difficult.

Finally, unpublished data (mainly our pilot studies for the development of psychophysical parameters in tDCS experiments), have shown that when the cognitive task is "slightly" above the threshold of difficulty, the effect of anodal stimulation online, but not offline, also affects accuracy.

I think the point raised by reviewer #1 is really relevant and would deserve a series of ad hoc investigations.

To mention this topic, we added a comment in discussion (line 244): “This latter finding supports a number of studies (for a review by Dedoncker et al. (2016), which suggest an effcacy of a-tDCS in reducing response times in the absence of any

of improved accuracy in cognitive tasks”.

  1. It is important to report all statistical effects (main and interaction) along with F and P values. Also, the authors must report a measure of effect size. 

REPLY: All F and p values have been provided in the results session; also, effect size (η2) has been added.

  1. The authors must report error bars in their figures. They are missing from all plots. 

REPLY: Error bars have been added in all figures of the manuscript

  1. Figures - Please use similar y-axis scale for sub-plots within a figure.

REPLY: We have made the y-axis consistent in figure 3. Moreover, to make the y-axis more similar even in figure 4, the minimum value has been set to 500 ms (as in fig 2). It has not been possible to adapt the maximum value of figure 4, because the vRTs in the tactile task were considerably slower (over 900ms).

  1. Please clearly define reaction time.

REPLY: Thank you for noticing this typo (line 142). Actually, we have used the term “response time” rather than “reaction time” throughout the text, as it is more appropriate to the paradigm type.

The definition of response time was provided on line 176, as follows: “Response time refers to the time between perceiving a stimulus and responding to a specific question about it. It is usually measured in milliseconds”.

  1. Please justify the stimulation montage used in this study.

REPLY: Usually, in studies that use parietal areas as target regions, the reference electrode is placed in the anterior portion of the hemisphere opposite to that of stimulation. This montage

ensure adequate stimulation in which most of the current reaches the target parietal region. In addition, it respects the safety rules required for the use of tDCS and allows to optimize the duration and density of the current delivered. Finally, this classic bicephalic mounting and electrode arrangement appears to be the most effective in cognitive task paradigms (see for example recent studies by; Marquardt et al., 2021; Martin, 2020; Hornburger et al., 2019; Vignali et al., 2019).

With respect to this topic we have added the following sentence (line 165): “This electrodes displacement ensures adequate stimulation in which most of the current reaches the target parietal region (namely rAG) and it respects the safety rules required for the use of tDCS”.

  1. Did the authors randomize the conditions to rule out any order effect?

REPLY: Yes. Thank you for pointing out this oversight. We have added the following sentence:

“The administration of the conditions was balanced across subjects” (line 105).

Brief bibliography provided in responses to reviewer 1

Dedoncker J, Brunoni AR, Baeken C, Vanderhasselt MA. A Systematic Review and Meta-Analysis of the Effects of Transcranial Direct Current Stimulation (tDCS) Over the Dorsolateral Prefrontal Cortex in Healthy and Neuropsychiatric Samples: Influence of Stimulation Parameters. Brain Stimul. 2016 Jul-Aug;9(4):501-17. doi: 10.1016/j.brs.2016.04.006. Epub 2016 Apr 12. PMID: 27160468.

Vignali L, Hawelka S, Hutzler F, Richlan F. No Effect of cathodal tDCS of the posterior parietal cortex on parafoveal preprocessing of words. Neurosci Lett. 2019 Jul 13;705:219-226. doi: 10.1016/j.neulet.2019.05.003. Epub 2019 May 4. PMID: 31063793.

Marquardt L, Kusztrits I, Craven AR, Hugdahl K, Specht K, Hirnstein M. A multimodal study of the effects of tDCS on dorsolateral prefrontal and temporo-parietal areas during dichotic listening. Eur J Neurosci. 2021 Jan;53(2):449-459. doi: 10.1111/ejn.14932. Epub 2020 Aug 25. PMID: 32746504.

Martin AK, Kessler K, Cooke S, Huang J, Meinzer M. The Right Temporoparietal Junction Is Causally Associated with Embodied Perspective-taking. J Neurosci. 2020 Apr 8;40(15):3089-3095. doi: 10.1523/JNEUROSCI.2637-19.2020. Epub 2020 Mar 4. PMID: 32132264; PMCID: PMC7141886.

Hornburger H, Nguemeni C, Odorfer T, Zeller D. Modulation of the rubber hand illusion by transcranial direct current stimulation over the contralateral somatosensory cortex. Neuropsychologia. 2019 Aug;131:353-359. doi: 10.1016/j.neuropsychologia.2019.05.008. Epub 2019 May 9. PMID: 31078549.

Reviewer 2 Report

In this study, Spitoni and colleagues investigated the metric component of body representation. Several studies have suggested that the right angular gyrus is involved in the tactile metric component; in the current study, Spitoni et al investigated whether the right angular gyrus is also involved in the visual metric component. Furthermore, they aimed to replicate previous findings on the tactile component. Their results confirmed the role of the right AG in tactile metricity, and supported the idea that the right AG is also involved in visual metricity. The aims and experimental design are clear and straightforward. However, there are several points in the introduction and discussion that will have to be more clearly explained. I also feel some details in the methods are missing and I have some suggestions to make the methods and results more complete and clear.

  1. In the introduction, the authors write „What both studies seem to suggest is that the distance comparison between two touched points across different body parts cannot rely on space-specific skin receptors.“ It is not entirely clear to me how this conclusion can be drawn from these studies, and I suggest the authors explain this more.
  2. Anodal tDCS is thought to have an excitatory effect, in contrast with cathodal tDCS which is thought to reduce neuronal activity. The current study does not explain this difference, which I think should be added for clarification. Furthermore, why did the authors decide to use anodal and not cathodal tDCS? I believe a short explanation of the rationale should be included.
  3. The task was to say in which condition the distance between the two dots was larger. How did participants have to report their judgment? Later on the phrase „vocal response times“ is used, so I‘m assuming that participants had to verbally report their judgment. How were response times exactly measured?
  4. Did participants have to put their sleeves up, so that the dots could be projected directly on the bare skin of the forearm? Or were the dots projected on the sleeves of the clothes participants were wearing?
  5. The authors write that 70 trials were obtained in each condition, and a total of 560 trials. However, from the description I arrive at a different number of trials. If there were 10 trials for each difficulty level, and there were 7 difficulty levels, there are indeed 70 trials per condition. But wouldn‘t that mean that there are 280 trials in total (2 conditions and 2 tDCS sessions)? Please correct me if I‘m wrong.
  6. From the part „Task and stimuli“ it is unclear whether the stimuli are presented at the same time, or sequentially. This only becomes clear when the reader arrives at figure 2, where the paradigm and its time line is illustrated. I would suggest to add this information (that the stimuli were presented sequentially with a certain ISI) in the paragraph „Task and stimuli“.
  7. Regarding figure 1: According to part A displaying the lasers, the distance between the dots on the forearm is smaller than on the hand, but on the picture below (part B) it's the reverse. This is a bit confusing; it would be better if both parts of the figure display the same example trial.
  8. TDCS was applied during the task for 15 minutes. Did participants manage to do the whole task within those 15 minutes?
  9. Line 202/203: „with faster responses to easy (596.5 ms) than medium (676.7 ms) and difficult judgments (722.1 ms)“. Were t-tests performed for this statement? If so, please provide the statistics.
  10. In the conclusion of experiment 1 (lines 215-217), the authors mention that stimulation of the right AG affects both the body and desk tasks. This is true only for the RT results, since accuracy was not affected by stimulation. I would add this to the conclusion to be more clear.
  11. Figure 3: please include error bars, and a description of the number of stars in the caption. Also, a figure for the accuracy results is missing.
  12. In the discussion, the authors speculate about the finding that tDCS influenced performance in both the body and desk task. Their discussion here is not that easy to follow and I would suggest to potentially rephrase certain parts to make it clearer. For example, the authors say that there are 2 possible explanations. They first mention the potential supramodal nature of the angular gyrus, and then write „On the other side,...“, which sounds like what follows will contradict this supramodal idea. However the part that follows about representation of numbers is actually meant to support the supramodal idea.
  13. Furthermore, I am not sure whether the finding that performance was affected in both the body and desk task shows that there is a supramodal function of the AG. After all, the sensory modality was visual in both tasks. I understand it shows that the AG is involved in events happening both on the body and in peripersonal space, but the authors should explain better how this can be considered supramodal.
  14. The second possibility that the authors mention as an explanation, namely that the task took place in peripersonal space, does not have any citations. I suggest that the authors refer to the literature more concretely to make their point.
  15. Lines 311-316: The paragraph starts with „With respect to accuracy...“, but the paragraph does not talk about accuracy but about reaction time instead.
  16. Do the authors have an explanation for the fact that stimulation seemed to only affect RT and not accuracy?
  17. Regarding the finding that performance was faster and more accurate for the desk vs the body task: could it be that the dots were simply better visible on the flat and smooth surface of the desk, compared with the curved and potentially hairy surface of the arm/hand?

Minor grammatical issues/typos:

In the caption of figure 1, it is written that the ISI is 50ms, however in figure 2 the ISI is portrayed as 100ms.

Line 123 „(insert technical features here)“ → looks like the authors still need to add these features

Line 129/130 „Participants were instructed to say in which condition (forearm/hand, or up/down) the distance between the two dots were higher.“ → the distance „was“, not „were“, higher. Also I wouldn‘t refer to distance as high or low, but as long and short or large and small. In this context, I would suggest to use large and small, so „the distance between the two dots was larger“.

Line 206: „a significant difficulty by task“ → I believe the word „interaction“ is missing.

Line 207: „with slower RTs in the easier trials on the desk“. According to the figure, RTs were actually faster in this condition.

Figure 4: the legend says „anodic“ instead of „anodal“

Line 286: „… that he stimulation“-→ „… that the stimulation“

Author Response

The authors would like to thank the 3 reviewers for the careful analysis of the draft and the time spent. Each point raised is appropriate and has helped to improve the manuscript. We hope we have interpreted the questions correctly

Below the replies provided point by point. (when provided, the mention to the number of lines refers to the revised version of the manuscript and NOT to the first version)

In this study, Spitoni and colleagues investigated the metric component of body representation. Several studies have suggested that the right angular gyrus is involved in the tactile metric component; in the current study, Spitoni et al investigated whether the right angular gyrus is also involved in the visual metric component. Furthermore, they aimed to replicate previous findings on the tactile component. Their results confirmed the role of the right AG in tactile metricity, and supported the idea that the right AG is also involved in visual metricity. The aims and experimental design are clear and straightforward. However, there are several points in the introduction and discussion that will have to be more clearly explained. I also feel some details in the methods are missing and I have some suggestions to make the methods and results more complete and clear.

  1. In the introduction, the authors write “What both studies seem to suggest is that the distance comparison between two touched points across different body parts cannot rely on space-specific skin receptors”. It is not entirely clear to me how this conclusion can be drawn from these studies, and I suggest the authors explain this more.

REPLY: Admittedly, reviewer 2 is right; it is not possible to draw this conclusion from the cited studies. What we meant was that since there are no specific cells that code for metric space on the body, one must refer to a structure that can process a function and not, merely, an intrinsic quality of the stimulus.

The topic was redrafted as follows: (line 70) “What both studies seem to suggest is that in order to perform a distance comparison between two touched points across different body parts we should hypothesizing a touch sensitive system (i.e., a mental body representation) which represents the state of the body rather than the physical features of the touching stimuli”.

  1. Anodal tDCS is thought to have an excitatory effect, in contrast with cathodal tDCS which is thought to reduce neuronal activity. The current study does not explain this difference, which I think should be added for clarification. Furthermore, why did the authors decide to use anodal and not cathodal tDCS? I believe a short explanation of the rationale should be included.

REPLY: This study can be conceptualized as a continuum with the other two previous studies in which anodal stimulation was used to modulate rAG activity in a tactile distance discrimination task. In this work, the same experimental design was repeated, but in a visual modality. Therefore, the type of stimulation, anodal, remained unchanged. As suggested we have added a short paragraph explaining the rationale, as follows: (line 87)“As this study can be conceptualized as a continuum with the other two previous studies, we used anodal and sham stimulation in all conditions”.

  1. The task was to say in which condition the distance between the two dots was larger. How did participants have to report their judgment? Later on the phrase „vocal response times“ is used, so I‘m assuming that participants had to verbally report their judgment. How were response times exactly measured?

REPLY: The subjects responded vocally; vocal response times were recorded through a microphone connected to the setup that administered the stimuli. The microphone was placed on the collar of the shirt, with a fabric adhesive.

  1. Did participants have to put their sleeves up, so that the dots could be projected directly on the bare skin of the forearm? Or were the dots projected on the sleeves of the clothes participants were wearing?

REPLY: laser points were projected onto the skin of the forearm

  1. The authors write that 70 trials were obtained in each condition, and a total of 560 trials. However, from the description I arrive at a different number of trials. If there were 10 trials for each difficulty level, and there were 7 difficulty levels, there are indeed 70 trials per condition. But wouldn‘t that mean that there are 280 trials in total (2 conditions and 2 tDCS sessions)? Please correct me if I‘m wrong.

REPLY: Ihe observation is correct and we thank you for pointing it out. We have double-checked the data set and realized that there was a typo. The trials were 20 (not 10) for each difficulty level.

The error has been corrected.

  1. From the part „Task and stimuli“ it is unclear whether the stimuli are presented at the same time, or sequentially. This only becomes clear when the reader arrives at figure 2, where the paradigm and its time line is illustrated. I would suggest to add this information (that the stimuli were presented sequentially with a certain ISI) in the paragraph „Task and stimuli“.

REPLY: We agree. The correction has been made.

  1. Regarding figure 1: According to part A displaying the lasers, the distance between the dots on the forearm is smaller than on the hand, but on the picture below (part B) it's the reverse. This is a bit confusing; it would be better if both parts of the figure display the same example trial.

REPLY: this is a very good point. we think the figures should be as clear as possible, so the point raised is very important. the figure has been modified as suggested.

  1. Line 202/203: „with faster responses to easy (596.5 ms) than medium (676.7 ms) and difficult judgments (722.1 ms)“. Were t-tests performed for this statement? If so, please provide the statistics.

REPLY: We did not perform t-tests as we used post-hoc. In fact, we run a ANOVA design with task (Body and Desk), stimulation type (anodal or sham), and difficulty (easy, medium, difficult) as within-subject factors and Bonferroni post hoc tests corrected for multiple comparisons (p < 0.05) were conducted. Usually post-hoc statistics are not reported in order not to " burden" the text. However, if the reviewer considers it useful, we can add a table with all the values in the supplementary session.

  1. In the conclusion of experiment 1 (lines 215-217), the authors mention that stimulation of the right AG affects both the body and desk tasks. This is true only for the RT results, since accuracy was not affected by stimulation. I would add this to the conclusion to be more clear.

REPLY: We agree. The sentence has been reformulated as follows: “Additionally, stimulation of the rAG affects vRTs both in the body and desk tasks”.

A similar point was highlighted by reviewer 1 and in agreement with both comments (reviewer 1 and reviewer 2) we added this topic to the discussion (line 244).

Anyway this point is very interesting, so we want to clarify what we think. The impact of tdcs on reaction time but not on accuracy is an effect that has been evidenced in several studies. A meta-analysis by Dedoncker et al. (2016), described that following a-tDCS over anterior regions, healthy subjects responded faster, while neuropsychiatric patients responded more accurate. Also, in a previous study by our group (Spitoni et al., 2013), we found that off-line a- tDCS over right angular gyrus improved vocal RTs but not accuracy. The reasons for such dissociation can only be hypothesized. On the one hand it is possible to imagine that some parameters such as the duration and, above all, the density of stimulation may play a role in modulating the facilitative effect of a-tDCS on accuracy. From another perspective, more speculative, it seems that in cognitive tasks, the effect of tDCS on accuracy emerges especially when the task is very difficult. In this framework it is not surprising that in the meta-analysis of Dedoncker and collaborators, an effect on accuracy emerges only in patients who, evidently, because of cognitive deficits, perceive tasks as more difficult. Finally, unpublished data (mainly our pilot studies for the development of psychophysical parameters in tDCS experiments), have shown that when the cognitive task is "slightly" above the threshold of difficulty, the effect of anodal stimulation online, but not offline, also affects accuracy.

  1. Figure 3: please include error bars, and a description of the number of stars in the caption. Also, a figure for the accuracy results is missing.

REPLY: Error bars were added. Regarding accuracy, we had thought of inserting an additional figure for it, but since the figure is easily guessed from the text (better performance in easy 80% vs. difficult 72% trials) the image would seem superfluous. Nevertheless, if the reviewer considers it essential, we will add a figure with the accuracy data, in the supplementary material.

  1. In the discussion, the authors speculate about the finding that tDCS influenced performance in both the body and desk task. Their discussion here is not that easy to follow and I would suggest to potentially rephrase certain parts to make it clearer. For example, the authors say that there are 2 possible explanations. They first mention the potential supramodal nature of the angular gyrus, and then write „On the other side,...“, which sounds like what follows will contradict this supramodal idea. However the part that follows about representation of numbers is actually meant to support the supramodal idea.

REPLY: the sentence has been modified as follows: “Regarding the second explanation several studies suggested a strong involvement of the left AG in number processing, such as digits subtraction and number”

  1. Furthermore, I am not sure whether the finding that performance was affected in both the body and desk task shows that there is a supramodal function of the AG. After all, the sensory modality was visual in both tasks. I understand it shows that the AG is involved in events happening both on the body and in peripersonal space, but the authors should explain better how this can be considered supramodal.

REPLY: We are not sure if we caught the sense of the question. We try to explain and apologize if we misinterpreted the comment. the concept of supramodal structure, is based on the possibility that the rAG is able to process the discrimination of distances in both visual (experiment 1) and tactile (2) modality. Therefore, there are two sensory modalities: vision and touch.

  1. The second possibility that the authors mention as an explanation, namely that the task took place in peripersonal space, does not have any citations. I suggest that the authors refer to the literature more concretely to make their point.

REPLY: Bibliographical references on peripersonal space have been added

  1. Lines 311-316: The paragraph starts with „With respect to accuracy...“, but the paragraph does not talk about accuracy but about reaction time instead.

REPLY: The paragraph on accuracy is linked to the entire paragraph on interpreting the main effect of the DESK condition; in fact, as reported in line 321, the data showed that responses on the desk were faster and more accurate than those on the body. To give greater flow and consistency we have modified the formal structure of the paragraphs as follows: “Another finding from experiment 1 is that responses on the desk were faster and more accurate than those on the body. With respect to speed, this may be because the body task requires additional processing such as retrieving an intrinsic property of the stimulus from the contingent properties of its contact with the skin. Regarding the latter interpretation, we could therefore hypothesize that only in the body task is a mental body representation recruited. With respect to accuracy, we know that in human life, actions are typically guided by the vision of the external environment. For example, every time we interact with an object we have to represent its visual characteristics, such as shape, size, orientation, temperature. So, the experience of evaluating quantities outside of our body is certainly more frequent than evaluating distances on the body. Therefore, we could suggest that the performance on the desk is faster and more accurate because of the effect of "practice and experience”.

  1. Do the authors have an explanation for the fact that stimulation seemed to only affect RT and not accuracy?

REPLY: This comment is related to a previous point. Please refer to the answer given in question 9.

  1. Regarding the finding that performance was faster and more accurate for the desk vs the body task: could it be that the dots were simply better visible on the flat and smooth surface of the desk, compared with the curved and potentially hairy surface of the arm/hand?

REPLY: Thank you for this possible suggestion. However, we think we can rule it out since in the visual stimulus tuning study (described in full in the supplementary session) each subject was asked to provide a qualitative assessment of stimulus characteristics (visibility, clarity, size) on the two surfaces. Ratings were provided on a Visual Analogic Scale. From what we found, the perceived quality was similar on both surfaces.

As mentioned in the discussion, we think that the facilitation detected in the DESK condition depends on the higher frequency with which we are used to experience the external world. Of course, this is only one possible explanation. it would be interesting to implement an ad hoc study.

Minor grammatical issues/typos:

In the caption of figure 1, it is written that the ISI is 50ms, however in figure 2 the ISI is portrayed as 100ms.

CORRECTED

Line 123 „(insert technical features here)“ → looks like the authors still need to add these features

CORRECTED

Line 129/130 „Participants were instructed to say in which condition (forearm/hand, or up/down) the distance between the two dots were higher.“ → the distance „was“, not „were“, higher. Also I wouldn‘t refer to distance as high or low, but as long and short or large and small. In this context, I would suggest to use large and small, so „the distance between the two dots was larger“.

CORRECTED

Line 206: „a significant difficulty by task“ → I believe the word „interaction“ is missing.

CORRECTED

Line 207: „with slower RTs in the easier trials on the desk“. According to the figure, RTs were actually faster in this condition.

CORRECTED

Figure 4: the legend says „anodic“ instead of „anodal“

CORRECTED

Line 286: „… that he stimulation“-→ „… that the stimulation“

CORRECTED

Reviewer 3 Report

This is a well written paper of results of elegantly executed experiments. What would be of some use is some physiological information - such as conduction times of stimuli when they are imposed on areas of the body at some distance away from the brain. Another suggestion is to expand the study into measurement of activation albeit indirect using an imaging technique that does not interfere with the stimulus delivery device. For this, I suggest functional near infrared spectroscopy.

Author Response

The authors would like to thank the 3 reviewers for the careful analysis of the draft and the time spent. Each point raised is appropriate and has helped to improve the manuscript. We hope we have interpreted the questions correctly

Below the replies provided point by point. (when provided, the mention to the number of lines refers to the revised version of the manuscript and NOT to the first version)

This is a well written paper of results of elegantly executed experiments. What would be of some use is some physiological information - such as conduction times of stimuli when they are imposed on areas of the body at some distance away from the brain. Another suggestion is to expand the study into measurement of activation albeit indirect using an imaging technique that does not interfere with the stimulus delivery device. For this, I suggest functional near infrared spectroscopy.

Reply: We thank reviewer three for his approval of our study and for providing useful suggestions. For example, the possibility of measuring the conduction time from the touching stimulus to the brain, would provide a clearer measure of response times in both the tactile and visual task; to say it better, knowing the conduction time from skin to brain, it would be possible to study the time of response processing net of the time of exposure to the stimulus. Unfortunately, in our study these values were not recorded, indeed we consider the suggestion useful for future studies.

The second suggestion, to use NIRS, delighted us.

In fact, the possibility provided by this technique to focus on a well-defined area, in our case the rAG, will allow us to expand our previous fMRI work, even in visual modality. 

Round 2

Reviewer 2 Report

I thank the authors for their thorough revision. They have addressed all my points, and I only have some very small remarks left (apologies, I’m not trying to be nit-picky!) or answers to some open questions. These can easily be addressed in a minor revision, and I’m happy to support publication with these points addressed.

Regarding the very first point, thank you for rephrasing this part. I only noticed a small typo: “we should hypothesizing” should be “we should hypothesize”

Regarding figure 1: I see you have switched the description of the lasers, so that the distances match the distances shown on the arm below, that’s great. Just a small typo: it now says “forearm/down” on the right side, which I presume should be “forearm/up”.

Regarding my previous point “Line 202/203: „with faster responses to easy (596.5 ms) than medium (676.7 ms) and difficult judgments (722.1 ms)“. Were t-tests performed for this statement? If so, please provide the statistics.”: Thank you for clarifying. I overlooked that you have conducted Bonferroni posthoc tests, apologies for that. I understand you don’t want to burden the text, but I do think it would be good to report all statistics for completeness. This can indeed be done in the supplementary part if the authors prefer this.

Regarding a figure for accuracy results: thinking about this, I agree it may not be necessary to illustrate the effect with a figure, since the main effects are on RT. I guess it should be sufficient to leave this as it is, with the effects mentioned in the text but not in a figure.

Regarding the discussion of the effect that tDCS affected both the body and desk conditions: apologies if I didn’t express myself clearly. I absolutely agree that the rAG can be seen as a supramodal area, since your results show that it is involved in both the visual and the tactile task which is in line with the literature. What I meant was that I’m not sure how this can explain that stimulation of rAG affected both the body and the desk condition from experiment 1, since in this experiment the stimuli were always visual. Reading it again, I understand that you mean that rAG has a more general role in number comparison, regardless of modalities, numbers, and sizes, is that right? That sounds like a plausible argument, I just misunderstood because of the phrasing, apologies for that. In that case, the two paragraphs (lines 283-292 and 293-302) actually belong together as one explanation. I would thus not write them as separate paragraphs, but as one paragraph. Also, I would replace “Regarding the second explanation” with “Furthermore”, since this second part (lines 293-302) is actually part of your first explanation.

Author Response

Dear reviewer #2, thank you for the additional points that allow the paper further improvement. Thanks also for the typos suggestions; we really appreciated them indeed.

Below are the responses to the points raised

I thank the authors for their thorough revision. They have addressed all my points, and I only have some very small remarks left (apologies, I’m not trying to be nit-picky!) or answers to some open questions. These can easily be addressed in a minor revision, and I’m happy to support publication with these points addressed.

  • Regarding the very first point, thank you for rephrasing this part. I only noticed a small typo: “we should hypothesizing” should be “we should hypothesize”

REPLY: correction has been made (line 71)

  • Regarding figure 1: I see you have switched the description of the lasers, so that the distances match the distances shown on the arm below, that’s great. Just a small typo: it now says “forearm/down” on the right side, which I presume should be “forearm/up”.

REPLY: correction has been made (line 143)

  • Regarding my previous point“Line 202/203: „with faster responses to easy (596.5 ms) than medium (676.7 ms) and difficult judgments (722.1 ms)“. Were t-tests performed for this statement? If so, please provide the statistics.”: Thank you for clarifying. I overlooked that you have conducted Bonferroni posthoc tests, apologies for that. I understand you don’t want to burden the text, but I do think it would be good to report all

REPLY: As suggested by the reviewer, statistical values were reported in a table in the supplementary (table 1S-line 448). As it is known, the Bonferroni's post-hoc comparisons do not give values of standard distributions (such as Fischer's F or Student’s t) but values inherent in Mean Difference (I-J), Standard Error and the lower and upper Bonds of the Interval of Confidence (95%). As a rule (better to say as usual in the scientific community), all of these indicators should not be reported when writing up the results; for this reason, table 1S shows the means and all the p values of each comparisons.

  • Regarding a figure for accuracy results: thinking about this, I agree it may not be necessary to illustrate the effect with a figure, since the main effects are on RT. I guess it should be sufficient to leave this as it is, with the effects mentioned in the text but not in a figure.

REPLY: we are glad to share this view

  • Regarding the discussion of the effect that tDCS affected both the body and desk conditions: apologies if I didn’t express myself clearly. I absolutely agree that the rAG can be seen as a supramodal area, since your results show that it is involved in both the visual and the tactile task which is in line with the literature. What I meant was that I’m not sure how this can explain that stimulation of rAG affected both the body and the desk condition from experiment 1, since in this experiment the stimuli were always visual. Reading it again, I understand that you mean that rAG has a more general role in number comparison, regardless of modalities, numbers, and sizes, is that right? That sounds like a plausible argument, I just misunderstood because of the phrasing, apologies for that. In that case, the two paragraphs (lines 283-292 and 293-302) actually belong together as one explanation. I would thus not write them as separate paragraphs, but as one paragraph. Also, I would replace “Regarding the second explanation” with “Furthermore”, since this second part (lines 293-302) is actually part of your first explanation.

REPLY: correction has been made (line 293)